## Research Article

hazardous drinking; brief interventions; alcohol; mHealth; qualitative

**Corresponding author:**
Abhijit Nadkarni;
Email: abhijit.nadkarni@lshtm.ac.uk

# Experiences of a mobile phone delivered brief intervention for hazardous drinking: A qualitative study nested in the AMBIT trial from Goa, India

Danielle Fernandes[1] , Ethel D'Souza[1], Seema Sambari[1], Marimilha Pacheco[1], Joseline D'Souza[1], Richard Velleman[1,2], Urvita Bhatia[1] and Abhijit Nadkarni[1,3]

[1]Addictions and Related Research Group, Sangath, Porvorim, India; [2]Department of Psychology, University of Bath, Bath, UK and [3]Department of Population Health, London School of Hygiene & Tropical Medicine, London, UK

## Abstract

**Background:** This study explores the experiences of participants receiving a mobile-based brief intervention (BI) for hazardous drinking in India, to determine characteristics that influenced engagement and examine perceived reasons for change in alcohol consumption.
**Methods:** Semi-structured interviews were conducted with 10 adult hazardous drinkers who received a mobile-based BI in the intervention arm of a pilot randomised control trial. Data were coded through an iterative process and analysed using thematic analysis.
**Findings:** Study participants reported a positive experience, with factors such as customised intervention delivery and personal motivation facilitating their engagement. Participants reported a reduction in quantity and frequency of alcohol use. This was credited to the intervention, particularly, its provision of health-related information, goal-setting content and strategies to manage drinking. Apart from alcohol reduction, participants reported improvements in diet, lifestyle, wellbeing, and familial relations.
**Implication:** By providing a context to explain the impact of the intervention, the learnings from this study can be used to strengthen the implementation of mobile-based interventions. This study outlines the scope for further research in digital health, such as Internet-based health interventions, and incorporating digital interventions within the ambit of existing health care programmes.

## Impact statement

Despite the increasing burden of alcohol-related problems, access to appropriate care remains inadequate, especially in low- and middle-income countries. Access to evidence-based brief interventions (BIs) for hazardous drinking is limited because of health-systems-related challenges such as shortage of health professionals and conflicting demands on the time of healthcare workers. Through our study, we examined the acceptability and feasibility of delivering BI to hazardous drinkers using inexpensive and easily available mobile text messaging services. We found that text messaging enabled BI brief was acceptable to hazardous drinkers. Personal motivation facilitated engagement with the intervention while time constraints acted as a barrier. The key intervention strategies that were perceived to help reduce alcohol consumption included the health-related information and goal-setting. The use of inexpensive and easily accessible basic technology to deliver BI to people with hazardous drinking is a potentially scalable strategy to increase access to care for drinking problems, especially, in low resource settings.

## Introduction

Alcohol misuse has gained increasing global attention as a major public health concern, as evidenced by its inclusion in the sustainable development goals agenda (Collin and Casswell, 2016). With high-income countries (HICs) tightening regulations on alcohol sales and advertising (World Health Organization (WHO), 2019), the industry has responded by expanding its growth in low- and middle-income countries (LMICs) like India (Esser and Jernigan, 2015). Despite India's reputation as having a culture of abstinence, alcohol consumption and alcohol-related problems have seen a steady increase (Devaux and Sassi, 2015; Rathod et al., 2015).

Drinking patterns that already cause harm (such as harmful and dependent drinking) have a lower prevalence than hazardous alcohol use (alcohol consumption that increases risk of physical or psychological harm) in India (Rathod et al., 2015). However, public health policy and funding

in India disproportionately addresses the former (Rathod et al., 2016). Furthermore, the shortage of trained health professionals acts as a considerable barrier to treatment access (Hazarika, 2013), while the stigma associated with alcohol misuse limits treatment-seeking (Mattoo et al., 2015). Technology has been championed as an innovative strategy to overcome the treatment barriers that exist in resource-constrained settings. Digital interventions have a strong evidence base in HICs (Fowler et al., 2016) and preliminary evidence from several LMICs also points to their feasibility and efficacy (McCarthy et al., 2018; Harder et al., 2020).

India has a rapidly growing telecommunication sector with almost 75% of the population owning a mobile phone (Haenssgen, 2019). However, only one in four mobile phone owners has a smartphone (Haenssgen, 2019), with socio-economically disadvantaged and rural populations being less likely to own smartphones (Poushter, 2016). Furthermore, poor Internet connectivity in rural areas can present additional challenges to Internet-based innovations (Haenssgen, 2019). On the other hand, low-tech interventions employing short message services (SMSs) and interactive voice response (IVR) which can be accessed on basic, affordable mobile phones have significant potential due to their greater reach, low cost, and ability to overcome barriers of language and literacy. Prior research in the Indian context has demonstrated the efficacy of SMS/IVR to deliver interventions for a range of health concerns, such as mental health, tobacco cessation, and treatment adherence (Rodrigues et al., 2012; Chandra et al., 2014; Basu, 2020), but there is no evidence on such interventions for alcohol use. The AMBIT (Alcohol use disorders-Mobile-based Brief Intervention Treatment) project was conceptualised to cover this gap by developing and testing a brief intervention (BI) for hazardous drinkers in India that can be delivered over mobile phones.

The AMBIT intervention package was developed during the formative research phase (Nadkarni et al., 2021), and further tested through a case series where the intervention content and delivery mechanism were refined through an iterative process (submitted for peer review). The final intervention was evaluated through a pilot randomised control trial (RCT) which generated preliminary impact estimates and allowed for fine-tuning of procedures for a definitive RCT (Nadkarni et al., 2022). A qualitative study was nested within the RCT to explore participants' experiences, barriers and facilitators of a mobile-based intervention in the Indian context. Furthermore, this study intended to explore participants' perceptions of how the intervention influenced drinking outcomes, to complement the findings of the RCT. This is particularly relevant in light of the limited evidence on developing and implementing mHealth alcohol-reduction interventions that are acceptable and feasible in the Indian context, and limited understanding of what makes such interventions work (Fowler et al., 2016; Field et al., 2019; Ghosh et al., 2022).

## Materials and methods

### Setting

The study was conducted in Goa, a small coastal state in India, which has a population of approximately 1.4 million (Government of India, 2011). With a liberal culture and cheaper alcohol prices, Goa has a relatively high rate of risky alcohol use compared to the rest of the country (Pillai et al., 2013). Prior research among adult male drinkers in Goa has estimated that the prevalence of

hazardous drinking is approximately 15% in primary care (D'costa et al., 2007) and 21.3% among industrial workers (Silva et al., 2003).

### Study design

The AMBIT intervention was evaluated through a pilot study designed as a parallel three-arm single-blind individually randomised-controlled trial. The three arms of the trial consisted of (i) the mobile-based intervention (experimental condition), (ii) brief-intervention delivered by a lay health worker and (iii) leaflet with information on hazardous drinking and alcohol reduction. The trial was conducted among a sample of 74 participants. This qualitative study was nested within the mobile-based intervention arm of the trial.

### AMBIT intervention

The mobile-based intervention was delivered over a duration of 8 weeks. Participants could choose whether they preferred to receive the intervention through SMS or IVR calls. The SMS and IVR calls both consisted of the same content, with the difference being the delivery mode. Participants could also choose the days of the week and time of the day (morning or evening) to receive the intervention. Messages/IVR calls were sent twice or thrice a week, based on findings from the formative phase of the study (17) and input from participants in the case series (submitted for peer review). Participants were able to customise the language (English, Hindi, or Konkani – the vernacular in Goa) used in the messages/IVR calls.

The intervention used a 'push' message format, where information was delivered and participants were given an option to respond to messages. These responses were optional, with further intervention content being delivered irrespective of whether participants responded or not. This functionality was based on feedback of participants from the case series who received a highly interactive 'push and pull' format which necessitated responses from the participants. Participants did not appreciate being required to respond, and suggested removing this mandatory response feature (paper submitted for review).

Each week, the intervention focused on specific content areas related to reducing hazardous alcohol use. These content areas were derived from the formative research of the study, and are summarised in Table 1.

### Study sample

The pilot RCT participants were recruited from educational institutions (two rural and five urban colleges), workplaces (police stations and bus depots) and primary care (two community health centres). The decision to focus on these sites was made due to prior research in these settings indicating high rates of hazardous alcohol use (Silva et al., 2003; D'costa et al., 2007; Pillai et al., 2013). The participants from all sites were adult (≥18 years) hazardous drinkers, identified through a score of 8 to 15 on the Alcohol Use Disorders Identification Test (AUDIT) (Saunders et al., 1993).

While only male hazardous drinkers were recruited from workplaces and primary care settings, both male and female hazardous drinkers were recruited from educational institutions. The decision to prioritise recruitment of males was based on prior research from the Indian context which indicated higher hazardous drinking rates among adult males (Silva et al., 2003; Pillai et al., 2013; Rathod et al.,

**Table 1.** Description of intervention components

| Intervention component | Description |
| --- | --- |
| Safe drinking | Safe drinking tips. For example, pacing drinking |
| Self-awareness | Recommendations to assess their alcohol use habits, and increase self-motivation to change |
| Alcohol reduction | Encouraging drinkers to reduce alcohol intake |
| Goal-setting | Prompts to identify goals for making changes to drinking behaviour |
| Situational content | Identification of common situations involving alcohol consumption and how to respond in a healthy manner |
| Self-reflection | Questions that help them observe, assess and analyse drinking behaviour in the context of their life |
| Drinking management | Prevention, risk reduction and coping methods within the context of hazardous drinking |
| Risk management | How to avoid and strategise for specific risky drinking situations that may prevent them from reaching their drinking goals |
| Motivation | Encouragement to continue with healthy behaviour adoption and maintenance |
| Drinking alternatives | Information about healthy alternatives to drinking that can help to reduce drinking urges |
| Review | Assessment of drinking behaviour in previous week |
| Other resources | Information about supportive health resources available |
| Urge management | Skills to overcome drinking urges |
| Actionable feedback | Useful tips that influence future health behaviour |
| Maintenance and relapse prevention | Assignments to help them practise and maintain healthier drinking behaviours |
| Goal management | Support for setting and maintaining non-risky drinking goals |
| Check-in messages | Short check -in messages to ensure movement towards the desired drinking goal |

2015). However, emerging research among young adults shows the growing prevalence of alcohol use among young women (Verenkar and Vaz, 2018), hence, it was decided to extend the intervention to young women in educational institutions settings.

Convenience sampling was used to recruit RCT participants into the present qualitative study. All participants who received the mobile intervention (mobile-intervention arm of the trial) were approached when collecting outcome data (outcome data were collected 3 months after the individual entered the trial). These participants were invited to participate in a qualitative interview, and all those who consented were entered into the present qualitative study.

### Data collection and analysis

Data were collected through in-depth interviews administered after the outcome evaluation which was conducted 3 months after recruitment into the trial. The semi-structured interview guide (Appendix 1 of the Supplementary Material) was designed to focus on domains consistent with the study objectives. The interviews were conducted between March and June 2020, and each interview lasted an approximate duration of 30–40 min. As a result of the COVID19 pandemic, interviews were conducted via telephone calls. Interviews were conducted in Konkani (local vernacular) by researchers over the phone. All interviews were audio-recorded, transcribed and translated to English.

Thematic analysis was used to analyse the qualitative data using NVivo version 12 (Braun and Clarke, 2006). In the first stage of the analysis, two independent coders (DF and SS) familiarised themselves with the data and generated codes from the raw data. Based on initial codes emerging from the raw data and the research question, potential themes were identified. The initial draft codebook was developed jointly by the two coders using these themes. In the next step, an initial set of five interviews was coded, after which the coders discussed inconsistencies in codes, differences in

coding and identified additional codes. Based on these discussions, the codebook was revised, and this was used to code the remaining interviews. After the coding was complete, the authors examined the themes and narrative extracts within codes in relation to the research questions (acceptability of the content and delivery of the mobile-based intervention, barriers to engagement, impact on drinking habits). Throughout the analysis process, the codebook was reviewed by AN and UB, and revisions were made with the consensus of the research team.

### Ethics

Ethical approval was obtained from the institutional review board of the implementing institution (approval number AN_2017_026) and the ethics committee of the Indian Council of Medical Research. Informed consent was obtained from all participants and all data were held confidentially and securely.

### Results

A total of 25 participants (23 males and 2 females) received the mobile-based intervention, with the majority recruited from workplaces (n = 14) and educational institutions (n = 9), and only two participants recruited from primary care centres. Of these 25 participants receiving the intervention, 10 consented to the in-depth interviews. Thus, the sample (n = 10) in the present qualitative study included seven students from educational institutions and three employees from local workplaces. All participants were males, aged between 19 and 49 years. The 40% consent rate for participation in the nested qualitative study was largely due to the onset of the COVID-19 pandemic during data collection, with participants being occupied with other responsibilities, moving back to their hometown due to the lockdown or unable to participate in telephone interviews due to technical issues. There was also a lack of female representation

in the qualitative study which was reflective of the low number of females identified as hazardous drinkers during screening and the low number of female participants in the intervention.

Four major themes were outlined based on participant narratives and the research questions; (i) general feedback on their experience with the intervention, (ii) barriers and facilitators which influenced their engagement, (iii) perceived outcomes and the reasons for these changes and (iv) suggestions to improve the intervention.

## Feedback on intervention characteristics

The participants expressed appreciation of the content and objectives of the intervention. Participants also reported that they were comfortable with the flexible mode of interaction and amount of content.

### Degree of interaction

Participants appreciated the flexibility in interaction provided by the intervention, which was suitable to participants who had differing preferences. While some participants expressed the need to report their progress and provide feedback, others preferred to read the content without being required to respond. This confirmed the acceptability of using 'push' messages which provided the option to respond but also allowed participants to read without responding.

"*But the thing is that I usually do not respond, that is my natural instinct. I do not respond to messages at all. So, me… I was taking in your info; I was taking in the guidance that was given and it was perfect… I am going through it, and you are asking me for feedback in between but the thing is that I really do not know at that moment. So, I… then I would not respond.*" – Participant 1MR009 (Male, 19 years).

### Amount of content

The length, number and frequency of the messages/calls were deemed satisfactory by most participants, with one describing it as the 'optimal' length. Some participants warned that lengthy or numerous messages could prove an annoyance and deter engagement and emphasised the importance of concise content that gets the message across.

"*If you are going ahead with the pre-recorded calls* (IVR) *then the current length of the calls is pretty much okay. Because through a one-minute call you are actually able to explain what sort of steps I need to take and also on the same hand they are not very long duration. So yeah, I think whatever calls you currently have that's an optimal sort of a call.*" – Participant 1MR021 (Male, 23 years).

"*I was pretty comfortable reading it. Because as I said it was not too long or it was not too short also… You were not getting bored by reading lot of messages because it was not that big, it was pretty much compact and whatever needs to be said, it is said.*" – Participant 1MR009 (Male, 19 years).

## Barriers and facilitators

### Technical issues

Minor technical issues were reported by a small number of participants, such as receiving many messages all at once, or issues with their phones.

"*I don't know what… it was some network issue or something, I don't know but I received a chunk of messages together… Having a message once after like once… once in a break is fine. Because as I said I got a chunk lot of messages, so I did not go through that chunk at times.*" – Participant 1MR009 (Male, 19 years).

### Intervention platform facilitated engagement

Participants described the benefits of the mobile-based intervention platform which allowed intervention content to be accessed when convenient. This was particularly useful for participants experiencing time constraints.

"*Because physical interaction, word to word interactions is not possible every time. As sometimes we are busy with meetings or some other things. So afterwards we can read messages, reply and forward. This should happen.*" – Participant 1HD038 (Male, 44 years).

"*Whenever I was free during that time I took the calls* (IVR) *and I didn't have any problems but obviously when I didn't get the time then I didn't take the calls. So, when I used to hang up the calls, they used to call me again. So, I didn't have any problem with that because it is good that they were calling once again to give some health tips but ya that's more or less all about it.*" – Participant 1MR021 (Male, 23 years).

### Personalised delivery facilitated engagement

The personalisation of the intervention by allowing participants to select preferred timings and days for delivery of the intervention supported engagement. Receiving messages/calls at a convenient time helped participants maintain engagement with the intervention.

"*Yeah, I was comfortable because I had given a certain time, so during that time only they used to send me messages, when I used to be free. So, I was very comfortable texting back, replying back to the messages.*" – Participant 1MR006 (Male, 23 years).

## Perceptions of intervention outcomes

### Drinking outcomes

Participants reported a change in consumption of alcohol as well as changes in drinking-related behaviour. Most described a reduction in alcohol consumption, while a few others mentioned that they had abstained from drinking entirely. A number of participants also described a change in behaviours associated with their drinking, such as drinking without the intention to get intoxicated and engaging in alternate activities.

"*I am just like that. I don't want to drink that much, I am fine. I had this much, it's enough for me. Just drinking for the sake of drinking. Not because I want to get drunk. I am just drinking maybe because I like the taste of it. But not to get drunk… …I am not drinking to get drunk, that is the change.*" – Participant 1MR009 (Male, 19 years).

### Impact of the intervention

While discussing the alterations in their drinking patterns, several participants attributed the change to exposure to the mobile-based intervention and described the mechanisms by which this change occurred. The intervention gave participants an insight into the amount of alcohol they consume and its negative consequences. This awareness of the risks associated with their drinking was described as a powerful impetus for change.

*"Basically, reading them you know, I came to know that this is risky, it is harmful for health. So, I used to think overall, like I am doing the same thing. Am I affecting my health? Including my health, am I affecting the society, the people around me? So, I started thinking over it. This is a thing, and I should change it." – Participant 1MR006 (Male, 23 years).*

Participants also recalled the information they learned through the intervention such as safe drinking practices and goal setting and described how they employed these strategies to manage their drinking.

*"It gives us the points on how you can avoid drinking if you are with friends and if you are going to some party then how to avoid that and not to drink much and even if you are drinking then how to drink with moderation. So basically, the drinking tips they used to give." – Participant 1MR021 (Male, 23 years).*

### External factors influencing outcomes

Participants described external factors that acted in synergy with the intervention to achieve an improvement in drinking patterns. The nation-wide lockdown imposed due to the COVID-19 pandemic was described as a facilitator of change, as restrictions on movement and the sale of alcohol disrupted social gatherings and limited participants' access to alcohol. Some participants also experienced pressure from their families to reduce drinking which motivated their decision to manage their alcohol consumption. These external factors were generally described in relation to the intervention, where the intervention generated awareness and provided targeted strategies, whereas external factors acted as additional motivators to drive change. The role of such external factors that interacted with the intervention to generate outcomes, was considered important in contextualising the findings.

*"Due to lockdown it (alcohol) also was not available and friends were also not able to come, so totally it stopped for 2 to 2 & half months or so. But when I read the tips from messages that time, I managed my drinking like how to behave. But because of lockdown I could not go outside and did not meet my friends also and I don't have the habit of drinking alone. So … it remained that way." – Participant 1HD038 (Male, 44 years).*

*"My family members were fighting with me, they were saying that I am just drinking, don't go to work… so that is the reason I stopped completely. And I got your support and this support also. So, I stopped everything." – Participant 1HD040 (Male, 49 years).*

### Personal and lifestyle changes

Aside from the changes in drinking-related outcomes, some participants also reported improvements in their general wellbeing. This included reduced stress, healthier diet, and lifestyle patterns, and enhanced interpersonal relationships.

*"But, before when I used to drink, I used to not feel hungry and all. So, I used to skip my food, my lunch, dinner, etc. But after these things I started doing; I like my routine; my schedule was perfect." – Participant 1MR006 (Male, 23 years).*

### Suggestions
### Alternate digital platforms

Several participants suggested other digital platforms, such as WhatsApp, Facebook messenger and Instagram, to deliver the intervention. These messaging applications were deemed more convenient as they were already widely used to communicate. On the other hand, SMS messages and IVR calls were often associated with spam or advertisements and consequently ignored.

*"Because usually, people use WhatsApp. So, WhatsApp was a better mode… Because normally these days nobody goes to check their messages because many messages come through the company and everywhere." – Participant 1MR006 (Male, 23 years).*

Participants also described the potential to add multimedia content and interactive 'Instagram stories' as another benefit of these platforms. Some mentioned the financial cost of sending an SMS as a disadvantage, while the use of messaging platforms would not incur any additional expenses.

### Blended intervention

Some participants suggested adding an interactive component within the existing intervention, through phone calls or face-to-face conversations. Some spoke about their need to express themselves or seek clarifications, whereas a few participants mentioned that they are unaccustomed to a mobile platform and felt more comfortable with in-person contact.

*"Actually you know face to face conversation is better than sending messages, no one… no one is interested maybe. Like I used to read some of it." – Participant 1MS004 (Male, 21 years).*

### Adding follow-ups

A small number of participants suggested adding follow-ups to monitor their progress after the intervention.

*"I have stopped drinking completely so I will have to reply to say this right, how I stopped, what I did then you can ask me right? Yes, I wanted to send messages just for you to know about it." – Participant 1HD040 (Male, 49 years).*

### Expanding the intervention scope

There were also suggestions to address other issues related to drinking through the digitally delivered intervention. This included adding content for areas such as other substance use and improving interpersonal relationships. The importance of adopting a preventive approach to alcohol use with targeted advice for adolescents and their parents was also mentioned by some participants.

*Rather than giving a treatment, preventing it would be better. For example, educating the parents. Like what happens with the parents is that… some of the parents don't introduce their children to alcohol. They just say oh you should not drink it. But why? No one tells them." – Participant 1MR009 (Male, 19 years).*

### Discussion

This paper reports the findings of a qualitative investigation nested within a pilot RCT of a mobile-based BI for alcohol use. This study explored the mechanism by which the intervention influences alcohol-related outcomes and identified intervention characteristics that facilitate or impede engagement.

Although not powered to test effectiveness, the pilot RCT indicated the potential of the intervention in reducing drinking (Nadkarni et al., 2022). This was reflected in findings from the nested qualitative study where participants' perceptions of change included reductions in quantity and frequency of alcohol use. These outcomes were largely credited to the intervention, particularly, its

role in providing information about the health and social repercussions of drinking. The informative role was linked to enhancing motivation to adhere to the intervention. Other aspects of the intervention, such as goal setting and strategies to manage drinking and cravings, also featured as facilitators of positive change. Furthermore, participants reported a positive experience with the intervention, which suggests acceptability for a mobile-based intervention targeting alcohol use in India.

These findings are consistent with evidence on BIs for alcohol use in other contexts. Global research on BIs points to the importance of using health information to increase motivation and engagement with the intervention content (Signor et al., 2013; Longstaff et al., 2014; Davey et al., 2015; Soares and Vargas, 2020). This approach ties in with the Stages of Change Model which describes raising awareness about the causes and consequences of a health behaviour as a method to increase an individual's readiness to change their behaviour (Prochaska and Velicer, 1997; Connors et al., 2013). This is particularly relevant for hazardous drinkers who are not experiencing harmful consequences from their alcohol use and have less readiness to change (at the pre-contemplation or contemplation stage of change; Connors et al., 2013; Davey et al., 2015). Other 'active ingredients' in alcohol BIs that improve intervention outcomes are those that improve self-efficacy through goal setting and management strategies, thus assisting individuals to prepare and act on change (Connors et al., 2013; L'Engle et al., 2014; O'Donnell et al., 2014).

Apart from alcohol reduction, an interesting unintended outcome of the intervention was improvements in diet, lifestyle, wellbeing and familial relations. The interactions between these factors and alcohol use were also reflected in participants' suggestions to address other-drug-use and interpersonal conflict that may accompany or exacerbate alcohol misuse. This points to the interplay of alcohol use with a range of other health-related behaviours and risk factors which act both as causes and consequences of hazardous alcohol consumption. These findings are consistent with other research studies which suggest addressing a broad set of interrelated health co-morbidities to improve the efficacy of alcohol BIs (McCambridge and Saitz, 2017; Boumparis et al., 2019).

Our findings also highlight implications for further development and implementation of mobile-based BIs in developing countries, namely, the importance of personalisation of the intervention, and potential for adapting the delivery platform through high-technology or blended interventions.

Our primary learning from this study was the importance of allowing for customisation of the intervention delivery and structure. The varying, and at times contradicting, feedback from participants on intervention features that facilitated participation pointed to the need for interventions that can be adapted to specific user's needs. To address this issue, digital interventions increasingly employ artificial intelligence (AI) that can adapt their approach based on user responses and behaviour during the course of the intervention (Tuerk et al., 2019). While such sophisticated AI technologies may be unfeasible in developing economies, low-tech interventions can incorporate features that allow personalisation of the user's needs. For example, the AMBIT intervention allowed delivery timings and language to be customised based on the participants' stated preferences at the point of recruitment. During the preliminary testing phase of the intervention, it became apparent that apart from logistical adaptations, it is also important to account for individual preferences that influence a participant's experience of the intervention and consequently, their participation. Feedback from the formative phase of the study showed that

participants had varying preferences on the degree of interaction and length of messages. The intervention adopted a flexible approach to address this issue, where the push-method gave participants the option to reply to messages at certain points during the intervention, without necessitating interaction. Such low-tech solutions to adapt the intervention design can prove essential when scaling up, to meet the diverse needs of a wide range of participants.

Secondly, our findings indicated the scope to expand the intervention delivery platform and mechanism. While the present study emphasised low-technology interventions that can be accessed through basic mobile phones, several participants suggested the use of social messaging platforms such as WhatsApp and Facebook messenger. Such web-based interventions hold greater allure as they are more user-friendly allowing for more versatility in their design and incorporate attractive features like videos. This preference could also be a reflection of the Indian government's increasing investment in telecommunication and digital innovation (Mukhopadhyay and Mandal, 2019). Recent initiatives like 'Digital India' and 'Bharatnet' have led to greater penetration of Internet use with India accounting for one of the largest smartphone markets in the world (Maiti et al., 2020). Thus, there is immense potential to expand the scope of mHealth interventions using messaging platforms to target India's growing Internet-using demographic. Despite the rapid progress in digitalisation there remains a disparity in Internet access, with women, lower socioeconomic groups, and those in rural areas being at disadvantage (Maiti et al., 2020). Furthermore, young adults tend to be more comfortable with Internet-based applications, as compared to older adults (Maiti et al., 2020). The stark digital divide in India highlights the need to adapt the delivery platform of mHealth interventions based on the profile and need of the participant. The success of Internet-based digital interventions for mental health (Kanuri et al., 2020; Gonsalves et al., 2021) indicates that it has potential among urban-dwelling Indian youth; but it is important that interventions using much simpler SMS or IVR technology are also retained to allow access to such programmes by those without Internet- and smartphone-based applications.

Another potential avenue for adapting the delivery of mHealth interventions is through blended interventions. Participants spoke about the utility of interpersonal contact to seek clarifications, allow for unstructured interaction, or provide check-ins to monitor progress through the intervention. While incorporating such interpersonal communication is not a scalable strategy for a mHealth intervention, there is scope to add mobile-based interventions within existing clinical or psychosocial care services (e.g., as part of counselling provisions in universities and workplaces). This would enhance the performance of existing programmes, by allowing medical professionals to navigate limitations of high caseloads and time constraints (Quaglio et al., 2017; Lemley and Marsch, 2021). Blended interventions also provide an opportunity to address related health concerns that accompany alcohol use, particularly, when the service provider has limited expertise in handling such wide-ranging conditions (Muench, 2014). Thus, integrating mHealth interventions into the care model holds promise as a cost-effective strategy to target a broader range of health issues. This strategy has proved effective to reduce alcohol consumption in other LMICs (Harder et al., 2020), and has been employed to target mental health in India (Gonsalves et al., 2021).

Limitations of our study included the low variability in our study sample which were all men, predominantly college students and to a lesser extent, working professionals. The lack of females and participants from primary care reflected the lower participation

rates of these groups in the pilot trial. Hence, our study could not capture the perspectives of these populations, limiting the generalisability of our findings. However, it is important to reflect that lower participation of certain groups (females and working professionals) in the trial could be indicative of the stigma around alcohol use and seeking care for AUDs. This should be an added consideration for future studies designing recruitment strategies for alcohol-reduction interventions. Secondly, our data collection was impeded by the onset of the COVID-19 pandemic which required some interviews to be conducted over the phone. This lowered participation in the study due to network challenges for participants located in remote areas. Additionally, phone-based interviewing may have influenced rapport building and judgement of non-verbal cues, thus affecting the quality of the data.

The main strength of this paper is its scope as the first study to explore the experiences of participating in a mobile-based BI for alcohol use in the Indian context. Thus, by providing a context to explain the impact of the AMBIT intervention, the learnings from this study can be used to strengthen the implementation of mobile-based interventions.

## Conclusion

The findings from this study point towards the acceptability of mHealth interventions for hazardous drinking in India and provide recommendations for designing such interventions. An important insight was the need to develop interventions that can be customised to diverse needs and preferences in the target population. This study also outlines the scope for further research in digital health, such as Internet-based health interventions, and incorporating digital interventions within the ambit of existing health care programmes. Finally, this study and the larger AMBIT project provide insight on the factors influencing acceptability and implementation of mHealth interventions for alcohol use in similar contexts with underdeveloped AUD treatment systems.

**Open peer review.** To view the open peer review materials for this article, please visit http://doi.org/10.1017/gmh.2023.51.

**Supplementary material.** The supplementary material for this article can be found at https://doi.org/10.1017/gmh.2023.51.

**Data availability statement.** The data that support the findings of this study are available on request from the corresponding author, A.N.

**Author contribution.** A.N. was involved in conceptualisation, methodology, writing – review and editing, supervision and funding acquisition. D.F. was involved in supervision, investigation, project administration, formal analysis and writing original draft and editing. U.B. was involved in methodology, supervision and writing – review and editing. R.V. was involved in conceptualisation, funding acquisition and writing – review and editing. S.S. was involved in investigation, formal analysis and writing – review and editing. M.P. was involved in investigation and writing – review and editing. E.D. was involved in data curation and writing – review and editing. J.D. was involved in investigation and writing – review and editing.

**Financial support.** AMBIT was funded by a grant from the Department of Health and Social Care (DHSC), the Foreign, Commonwealth & Development Office (FCDO), the Medical Research Council (MRC) and Wellcome (Grant No. MR/P020348/1). The funders did not have any role in study design; in the collection, analysis and interpretation of data; in the writing of the report; and in the decision to submit the article for publication.

**Competing interest.** The authors have no competing interest to declare.

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
