## [Reviewer Report]

Professor Gary Belkin

Editor-in-Chief

Global Mental Health

Dear Prof. Belkin,

Experiences of a Mobile Phone Delivered Brief Intervention for Hazardous Drinking: A Qualitative Study nested in the AMBIT trial from Goa, India.

I am pleased to submit our paper describing the findings of a qualitative study nested in a feasibility trial of text messaging intervention for hazardous drinking in India. The contextually appropriate Brief Intervention (BI) was developed through a systematic process of intervention development described in separate peer-reviewed publications.

In our view, this paper represents critical intervention development research in global mental health for several reasons: first, the BI is developed using a systematic intervention development process, thus enhancing its contextual relevance; second, the BI is designed to be delivered using basic mobile phone technology available in low- and middle-income countries thus increasing its potential for scalability; and third, the BI is built around a theoretical orientation which has a strong grounding in the existing evidence base, which makes it relevant to a global audience.

We believe that these are the kinds of rare global mental health research outputs which have strong relevance even to high income countries where there are large treatment gaps and concerns about the dissemination of psychological treatments, and hence are suited for an international journal with a high-impact on the field such as the Global Mental Health.

We look forwards to your reply.

Regards,

Abhijit

---

## [Reviewer Report]

Thank you for submitting this paper. I enjoyed reading it and thought the intervention itself and the qualitative study were both interesting and congratulate the authors on such a great paper. The paper overall is well-structured and well-written. The methods used were appropriate and described well. My only comment here is that on pg. 11, more details of the ethical approval should be provided e.g. application no., for the purposes of transparency. Appendix A was not included so I was unable to comment on the interview guide.

The findings were clear and discussed within the context of the wider literature. In reporting qualitative researching findings, it is customary to include an interview number and/or participant number when providing direct quotes; this should be done here in the spirit of transparency.

There are a few minor spelling/grammar errors throughout, so the paper does need to be proof-read. Some of them I have picked up as follows:

Pg. 4, line 48 – data were, not data was; and throughout the paper

Pg. 10, line 186 – needs editing “and took and had”

Pg. 11, lines 215-217; repetition with the no. of participants and the breakdown of this

Pg. 18, line 406 & pg. 19, line 431 – participants’ not participant’s

You should also check some of the line spacing (due to ‘justify’ alignment)

---

## [Reviewer Report]

Well written paper and adds to the evidence of importance using low cost digital technologies like mobile phones to deliver BI to help quit hazardous drinking in low resource settings.

Will recommend following minor revisions before paper is published:

1. Line 186. Remove the words “and took”.

2. Line 169 mentions “Only male hazardous drinkers were recruited….”, and in Line 215, under the “Results” section, it is mentioned that 2 females were also recruited. Line 225 states that there was a lack of female representation due to the nature of study. Despite the inclusion criteria stated in Line 169 about recruiting only males, 2 females were still recruited. Line 500, it is stated that “study sample which were all men”, despite having 2 females in your study. Kindly explain this contradiction in the inclusion criteria of your study.

3. Same line is repeated twice. Line 169 “Of the 25 participants …received the intervention” and Line 216 “The 25 participants who … 2 females.”

4. Line 219. “The sample in the present … from local workplaces.” It states that 7 students were recruited from educational institutions, and in the previous line, it is stated that 9 participants were recruited from the educational institutions. One line states 14 participants were recruited from workplaces, and in the next line, it states 3 employees from local workplaces were recruited. The numbers do not match. Kindly explain the numbers completely so that everything adds up.

5. Line 47. Kindly rephrase this statement.

6. Line 367 - 370. Earlier in the paper, it is mentioned that “push” message format was used in the study. In the mentioned line, suddenly a social media platform is mentioned. This suggestion is not valid for the mode of the study through which it was conducted.

7. Bulk messages being sent through the medium of SMS is not allowed by TRAI. Yet, this study was conducted through it. Why did you choose this medium?

8. Line 376. Line to be modified into past-tense, i.e., “they were”.

The numbers mentioned in the paper do not match with what is written in pages above and below.